# Across-Track and Multi-Aperture InSAR for 3-D Glacier Velocity Estimation of the Siachen Glacier

**Vijay Kumar [1],\*, Kjell Arild Høgda [2] and Yngvar Larsen [2]**

1   School of Electronics Engineering, VIT Vellore, Vellore 632014, India
2   Norce, 6434 Tromsø, Norway; ynla@norceresearch.no (Y.L.); kjho@norceresearch.no (K.A.H.)
\*   Correspondence: vijaykumar@vit.ac.in

**Abstract:** Interferometric Synthetic Aperture Radar (InSAR) remote sensing generally lacks deformation sensitivity in the along-track direction. In this proposed approach, across-track observations from conventional InSAR, using both ascending and descending passes, were superimposed with the along-track movement derived from multi-aperture InSAR (MAI) to determine the full three-dimensional (3-D) velocity of the Siachen Glacier in the Karakoram range of the Himalayas. The along-track velocity signal is essential for estimating the movement component in the north/south direction, which is needed for a complete delineation of 3-D deformation. The velocity observed was improved using the MAI technique in comparison to the conventional ascending/descending 3-D velocity estimation approach, and substantial differences were noticed between these two methods, particularly in the lower part of the glacier, which is moving almost in an along-track (north/south) direction. Glacier velocity varied from 0.3 md$^{-1}$ in the accumulation zone to 0.60 md$^{-1}$ in the terminus zone of the Siachen Glacier using this newly proposed approach. This study presents a 3-D velocity estimation without any preconceived assumptions regarding the flow conditions of glaciers and without any azimuth ambiguity.

**Keywords:** SAR Interferometry; MAI; glacier; Himalaya; velocity

## 1. Introduction

Glacier flow measurements are fundamentally important for studying the mass balance and strain changes in glaciers and ice sheets [1–3] (Joughin et al., 1998; Gray et al., 2001; Rignot, 2002). Synthetic Aperture Radar Interferometry (InSAR) is a powerful geodetic tool for measuring a glacier's velocity and strain rate with high accuracy. Spaceborne repeat pass synthetic aperture radar (SAR) data acquired by remote sensing satellites ERS-1/2, Radarsat-1 and -2, ENVISAT, Advanced Land Observing Satellite (ALOS) Phased Array L-band SAR (PALSAR), PALSAR-2, TerraSAR-X, and Cosmo-SkyMed have been widely exploited to estimate the surface motion (velocity) of ice sheets and glaciers using the interferometric approach [4–7]. Most of glacier or ice sheet surface velocity studies have been carried out in polar regions using repeat pass SAR Interferometry or an intensity tracking approach [2,8,9]. Joughin et al., 1998 [1] demonstrated the potential of the InSAR technique to estimate the three-dimensional ice velocity in Greenland. A major limitation of InSAR-based velocity estimations is that this technique is only sensitive to two-dimensional motion along the line of sight (LOS) of SAR sensors. However, without incorporating north/south (along-track) velocity sensitivity, precise velocity cannot, in general, be estimated. Bechor and Zebker [10] demonstrated the InSAR approach for velocity estimation in the range direction (LOS) and the multi-aperture InSAR (MAI) approach for velocity component estimation in the azimuth (along-track) direction using a single InSAR pair of ERS-1/2 data. Gourmelen et al., 1998 [11] used InSAR and MAI for the 3-D velocity estimation of Icelandic glaciers. Gray 2011 [12] showed that multiple InSAR interferograms can be used to estimate LOS displacements in three different orientations to solve the full

3-D displacement of the glacier surface. This approach requires three InSAR pairs with three different orientations. MAI works with the amplitude and phase division components of return radar signals, and without these, the displacement signal in the azimuth direction cannot be obtained. Preserving the coherence between acquisitions is a precondition. More detail on this follows in the Methodology section.

Earth surface movement must be characterised using three-dimensional observations, and a lack of any single degree of freedom can mislead the 3-D velocity measurement [1]. Some systematic studies on Himalayan glaciers have been reported [6,13,14], but more realistic three-dimensional surface velocity studies using across- and along-track interferometry in the Himalayas are required.

In this study, two different approaches were used to estimate the 3-D surface movement of glaciers from radar LOS displacements. The first approach [1,6] estimated surface movement, assuming that the glacier's motion was parallel to the plane. In the second approach, north-south and south-north pass (NSSN) InSAR observations were combined with the azimuth velocity from MAI without any assumptions regarding glacier flow conditions. Along-track displacement was estimated from forward- and backward-looking interferograms, which were created by splitting the azimuth beam parallel to each of the (SLC) images [10,11,15–17]. Herein, we demonstrate a modified MAI technique with perfect azimuth error suppression for the precise velocity pattern estimation of the Siachen Glacier, and a comparison is made with the results from Joughin et al.'s 1998 [1] approach. ERS-1/2 tandem data in the NSSN pass were used to decipher the three velocity components without any assumption that the flow was parallel to the surface, which has been considered a prerequisite in earlier studies. Himalayan Glaciers preserve a high correlation of radar echoes for one day from the temporal baseline. Consequently, a high-density surface movement field could be obtained. The movement results were analysed by considering surficial and temporal variations and, for the consistency of the approach, it was compared with cross-correlation-based subpixel offset tracking [18,19] derived from velocity using the Landsat-8 (L-8) data pair 2017–2018.

## 2. Study Area and Data Sets Used

The Siachen Glacier, located in the Karakoram (K-2) range of the northwestern part of the high Himalayas [20] in the Jammu and Kashmir (J&K) state of India, constitutes a major source of water for the Indus River system. Compared to other glaciers in the Himalayas, Siachen is the largest valley glacier centred at a latitude of 35°20′N and longitude of E77°11′E (see Figure 1). This glacier is nourished by 11 small and large glaciers originating from the K-2 range of mountains. The upper part of this glacier and its tributaries form a vast ice field in the region. It feeds the Nubra River (also known as the Shaksgam River), which flows parallel to the Karakoram range. During the winter season, the average snowfall in this region is about 10.5 m. The air temperature varies from −10 °C to −50 °C [21], and the elevation varies from 3620 m at the snout to 5753 m near the zenith of the glacier.

To estimate the glacial flow pattern, we needed two SAR images of slightly different orbital geometries and a digital elevation model (DEM) from the same area for topographic phase correction. ERS-1 and -2 tandem data sets available in the European Space Agency (ESA) archive were used for this study. The details regarding ERS-1 and -2 tandem data sets used in this study are summarised in Table 1. Two interferometric pairs of data, one month apart in time difference, collected during the ERS tandem mission in ascending and descending passes with a one-day temporal difference were used to obtain the results. Seasonal snowfall effects could not be estimated with these limited data sets, but LOS motion using two passes can be used to resolve 3-D velocity by assuming that glacial velocity does not change during the acquisition of two pairs. The Shuttle Radar Topographic Mission (SRTM) DEM projected using radar coordinates was used to compute the topographic phase component. The height-simulated phase was subtracted from the InSAR phase to obtain the topographic-corrected phase equivalent to the displacement of terrain.

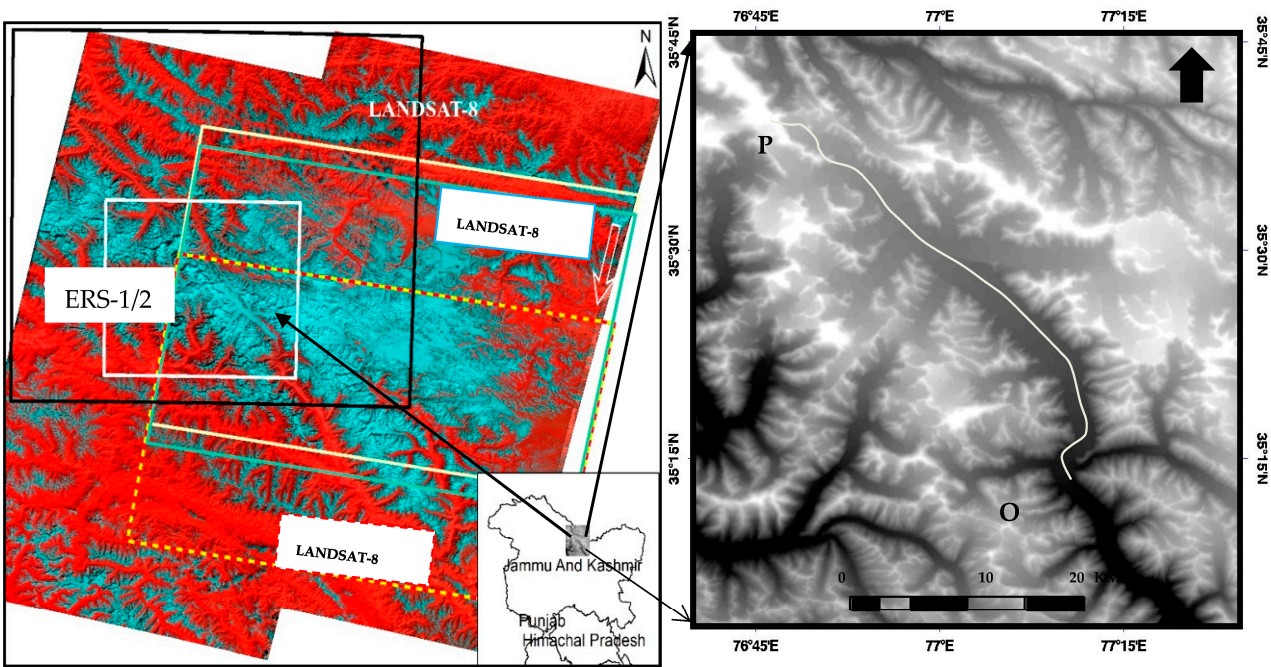

**Figure 1.** Location map of Siachen Glacier Left: various satellite sensor (optical and SAR) acquisition scenes covering the Siachen Glacier. Right: topography of local area and a line segment OP, starting from the terminus front at O to the accumulation zone P, drawn along the central portion of the glacier.

**Table 1.** ERS-1/2 tandem image pairs with satellite specifications.

| ERS1_ERS2 Orbit1_Orbit2 | Date1_Dat2 | PASS (A/D) | ⊥r BASELINE (m) |
|---|---|---|---|
| 24642_4969 | 01/04/1996_02/04/1996 | A<br>A | 110 |
| 25093_5420 | 02/05/1996_03/05/1996 | D<br>D | 114 |

## 3. Methodology

InSAR fringes were generated by multiplying the SAR signal with the complex conjugate of a signal acquired with a slightly different orbital geometry but with the same satellite track. In this way, the phase difference calculated between these two acquisitions is the sum of many components and is given by [22]:

$$\phi_{\text{Insar}} = \phi_{\text{def}} + \phi_{\text{topo}} + \phi_{\text{atm}} + \phi_{\text{orbit}} + \phi_{\text{noise}} \tag{1}$$

where $\phi_{\text{def}}$ is the deformation phase due to the displacement of the LOS during repeat SAR acquisitions. The topographic phase ($\phi_{\text{topo}}$) was calculated using a DEM. $\phi_{\text{orbit}}$ is a phase due to the incorrect knowledge of satellite orbits, $\phi_{\text{atm}}$ is a phase change due to different atmospheric delays between the acquisitions, and $\phi_{\text{noise}}$ is additive noise due to the variability in scattering from the pixel, SAR system thermal noise and co-registration errors.

Topographic correction techniques using DEM, as well as the three-pass approach, have been reported by [23,24]. Deramping the interferograms has been performed using orbital models for the ESA satellites ERS-1/2 [25]. Precise orbital models help to estimate the geometrical baseline and the removal of the orbital phase component from repeat pass interferograms. Fringes were flattened and then unwrapped using a statistical cost

network flow algorithm for phase unwrapping (SNAPHU) [26] where the phase, due to displacement of glaciers in the radar line of sight, is given as:

$$\phi_{defo} = \frac{4\pi}{\lambda}\Delta r \tag{2}$$

where $\Delta r$ is the LOS displacement between two SAR acquisitions.

Kwok and Fahnestock [27] combined radar observations from NSSN passes to quantify the 3-D movement vector by assuming that glacier flow is parallel to the surface. This technique can give an accurate estimation of velocity if glaciers are aligned to the ground range direction of SAR acquisition. If glaciers are oriented along the azimuth direction, then the accuracy of measurements is limited and cannot be interpreted as true velocity. Accordingly, there is a need for a technique that performs precise velocity estimation, which is independent of the orientation of the glacier flow direction. Herein, the azimuth direction velocity component is computed using the MAI approach, which is described in the following section.

Along-track InSAR has been in use for applications such as finding the velocity of on-ground objects and ocean currents [28]. Herein, along-orbit and back-looking interferograms are created by splitting SAR beams into two components in forward and backtrack directions, respectively, for a pair of SAR images. This method of interferogram generation is known as multi-aperture InSAR (MAI) [9,29]. Subtracting forward- and backward-looking interferograms makes the phase representative of azimuth displacement between the radar passes.

Typical radar geometry is shown in Figure 2. Here, the radar squint angle is $\theta_{sq}$, and the antenna angular beam width is $\alpha$. Forward- and backward-looking interferograms are formed using respective antenna beamwidths and integrated around $\theta_{sq} + \theta_f$ and $\theta_{sq} - \theta_b$, respectively.

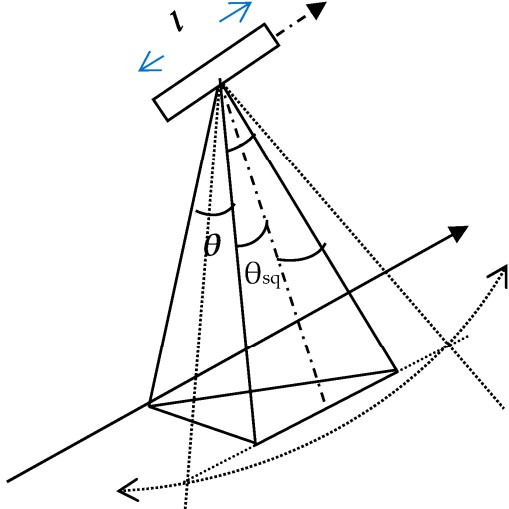

**Figure 2.** Along-track imaging geometry for forward- and backward-looking interferograms by squinting at the angle $\pm\beta$.

The displacement phase was obtained by subtracting forward- and backward-looking InSAR. The MAI phase, as a function of azimuth displacement x, the forward look squint angle $\theta_f$ and the backward look squint angle $\theta_b$, is given as:

$$\varphi_f = -\frac{4\pi}{\lambda}\sin(\theta_f) \quad \text{and}$$
$$\varphi_b = -\frac{4\pi}{\lambda}\sin(\theta_b) \tag{3}$$
$$\varphi_{MAI} = -\frac{4\pi}{\lambda}[\sin(\theta_f) - \sin(\theta_b)]$$

The relationship between squint angle $\theta_{sq}$ and the azimuth (Doppler) frequency $f_a$ is given by:

$$f_a = \frac{2}{\lambda}v_s.u_r = \frac{2}{\lambda}v_s\sin(\theta) \tag{4}$$

where $v_s$ is the satellite velocity vector $v_s = \|v_s\|$ , and $u_r$ is the unit velocity vector in the line of sight. Thus, the MAI phase can be given as:

$$\phi_{MAI} = -\frac{2\pi}{\lambda}(f_{a,forward} - f_{a,back})x \tag{5}$$

$$f_{a,forward} - f_{a,\ back} = 456.57$$

In the Siachen analysis, the Doppler difference between looks was $f_{a,forward} - f_{a,back} = 456.57$Hz, and $v_s$ = 7552.53 m/s, such that the effective sensitivity was about 16.5 m per fringe. Note that this is different from Bechor and Zebker [9], where the common Doppler bandwidth between the original two SLCs was assumed to be 100% of the PRF, leading to a sensitivity of 10 m per fringe. In practice, in order to suppress azimuth ghost effects, only about 80% of PRF is processed in the SLC. In addition, the common Doppler band was reduced by about 15% of PRF in this particular data set due to a Doppler mismatch between the two scenes. The effective Doppler difference between the midpoints of the two looks was then reduced to about 30% of the PRF instead of the assumed 50% of reported work [9]. Figure 3 shows the schematic flow of the generation of along-track displacement using either a descending or an ascending InSAR pair.

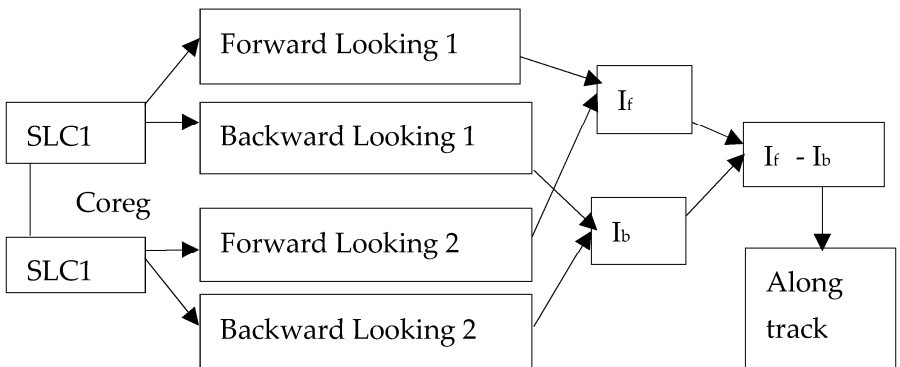

**Figure 3.** Flow of splitting two co-registered SLCs images for the generation of along-track displacement using MAI. $I_f$ and $I_b$ are forward- and backward-looking intreferogram phases, respectively.

*Estimation of 3-D Velocity Using Ascending, Descending InSAR and MAI (ADIMAI)*

The radar observations, taken from NSSN passes and along-track observations, were combined to resolve the true 3-D movement vectors. We introduced an acronym ADIMAI for ascending and descending pass InSAR and MAI. The following sections show how ADIMAI was used for complete 3-D velocity estimation involving three individual deformation components.

The midline points of the glacier were taken into consideration, while LOS velocities from NSSN pass InSAR pairs and MAI were used to estimate the 3-D velocity of the glacier along the central line. We chose to focus on the transect along the glacier, which is the unwrapped phase, with a manually chosen reference path to an off-glacier point that was assumed to be stationary. Figure 4 illustrates the motion components involved in estimating the 3-D velocity vector. The estimated LOS displacement using NSSN pass SAR pairs were $d_{aLOS}$ and $d_{dLOS}$, respectively. $D_{mai}$ is the along-track deformation component computed using MAI and applied on NSSN SAR pairs. X, Y, and Z are local rectangular

coordinate axes and i, j, and k are the unit vectors along them, respectively. Vector D is the 3-D deformation of the glacier and can be expressed as:

$$\vec{D} = X\vec{i} + Y\vec{j} + Z\vec{k} \tag{6}$$

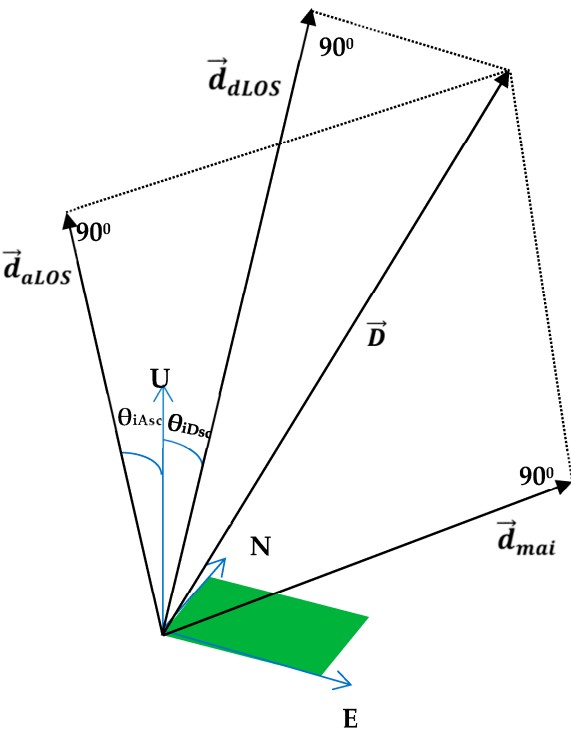

**Figure 4.** The velocity components from ascending pass ($d_{aLOS}$), descending pass ($d_{dLOS}$) and along-track ($d_{ati}$) interferometry techniques used to estimate three-dimensional glacier motions. D represents the three-dimensional motion in a local rectangular coordinate system.

The value of D was investigated using across- and along-track LOS deformation components. Known orbital state vectors of ERS-1/2 with high precision were used to estimate the unit vectors along an ascending pass LOS deformation ($d_{aLOS}$), descending pass LOS deformation ($d_{dLOS}$), and azimuth direction component ($d_{mai}$). Unit vectors along $d_{aLOS}$, $d_{dLOS}$, and $d_{mai}$ were defined as $\vec{u}_{aLOS}$, $\vec{u}_{dLOS}$, $\vec{u}_{mai}$, respectively. Estimated displacements can be expressed as a dot product of respective unit vectors with the true deformation and vector given as:

$$\vec{u}_{aLOS}.\vec{D} = d_{aLOS} \tag{7a}$$

$$\vec{u}_{dLOS}.\vec{D} = d_{dLOS} \tag{7b}$$

$$\vec{u}_{mai}.\vec{D} = d_{ati} \tag{7c}$$

in LOS because of ascending, descending passes, and MAI, respectively. Here $d_{ati}$ is a result of components $d^a_{ati}$ and $d^d_{atiD}$ from ascending and descending pass InSAR pairs, respectively [30,31].

The system of equations was solved for X, Y, and Z components of $\vec{D}$ using the matrix inversion of a product of matrices, such as:

$$\begin{pmatrix} \vec{u}_{aLOS} \\ \vec{u}_{dLOS} \\ \vec{u}_{mai} \end{pmatrix} \cdot \begin{pmatrix} X \\ Y \\ Z \end{pmatrix} = \begin{pmatrix} d_{aLOS} \\ d_{dLOS} \\ d_{mai} \end{pmatrix} \tag{8}$$

With the availability of one InSAR LOS and two InSAR LOS displacements, one degree of freedom and two degrees of freedom could be resolved in the estimated flow by assuming that the flow was parallel to the surface. Here, with MAI, three InSAR movement components were available- hence, three degrees of freedom were resolved without any assumption about the flow conditions of the glacier. Figure 5 shows the flow chart of the methodology adopted.

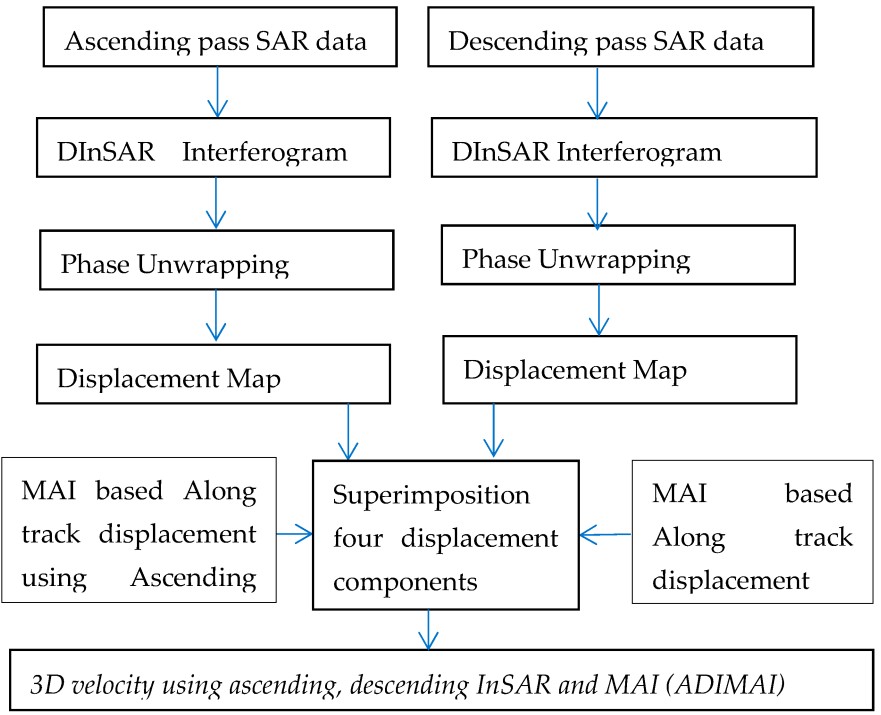

**Figure 5.** Flow chart for 3-D movement estimation using ascending and descending pass and corresponding MAI-based components.

## 4. Results and Discussion

The ADIMAI approach was used to study the Siachen Glacier flow estimation. The results obtained using available data sets (see Table 1) were best presented as velocity fields along a profile following the central line of the glacier. LOS velocity fields using NSSN pass InSAR pairs are shown in Figure 6a,b, respectively, with colour shading along the midline of the glacier representing velocity variation.

The change in geometry for the two passes was responsible for the different estimated velocity values during a one-month time difference. It was observed that the velocity increased with altitude and reached a maximum where tributary glaciers met the upper part of the Siachen Glacier.

At points B and C in Figure 6a, the largest speed was noticed due to two tributary glaciers from both sides of the Siachen Glacier feeding into the glacial mass. Glacier section OA (Figure 6a) is perpendicular to the radar ground range, and the movement component from MAI is the largest in this section. Figure 7 shows the variation in the along-track component derived from descending InSAR pairs using MAI. It was observed

that wherever the glacier turned toward a northerly direction, the along-track velocity component increased.

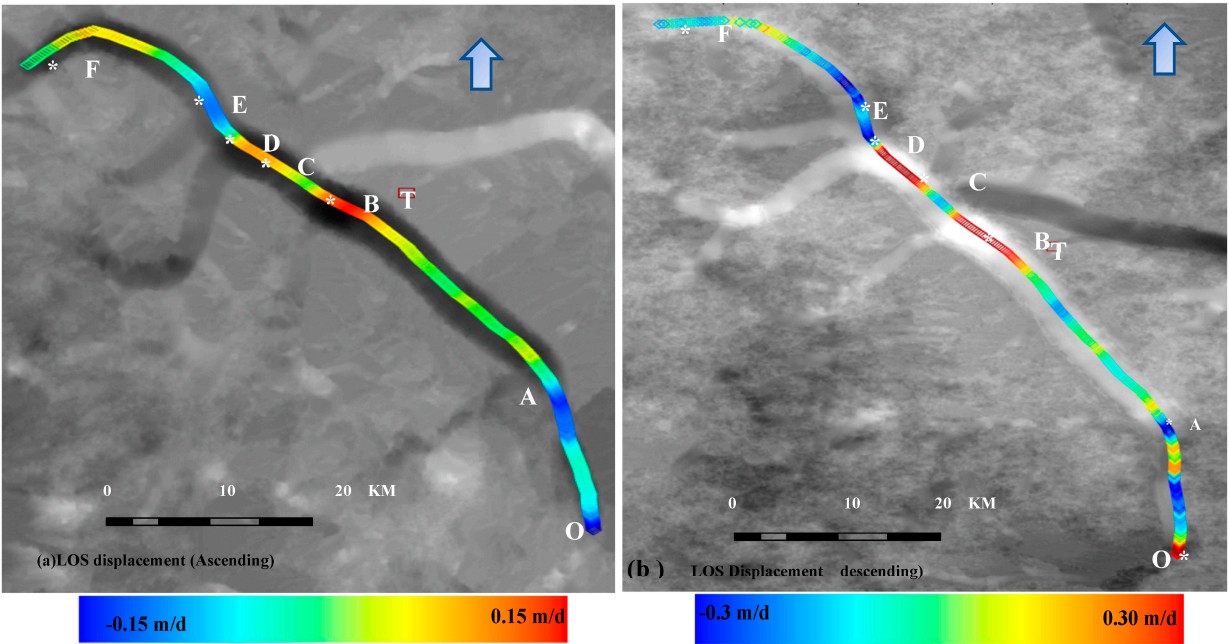

**Figure 6.** Velocity fields using (**a**) Ascending pass InSAR pair of 1–2 April 1996. (**b**) Descending pass InSAR pair of 2–3 May 1996. Coloured (gray) scale bar indicates LOS movement along the central line of the glacier. * is demarcation point.

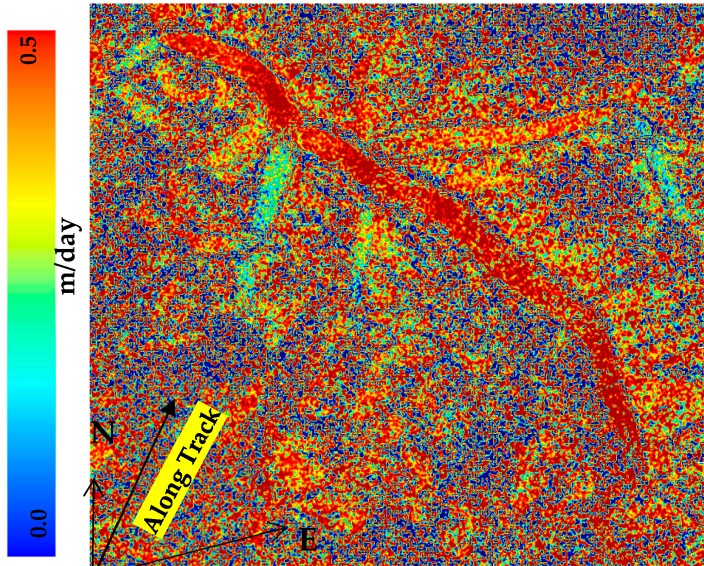

**Figure 7.** Along-track differential MAI from descending InSAR pairs.

Figure 8 shows four plots (i), (ii), (iii), and (v) reprinting velocity along the central line OF (in Figure 6a) derived from ADIMAI: only descending pass MAI, proxy 3-D derived using descending pass InSAR and ascending pass InSAR pairs using the SAR pair of 2 May 1996 and 3 May 1996 Landsat-8-derived, respectively. Plot (iv) on Figure 8 shows altitude variation along the central line OF on the glacier.

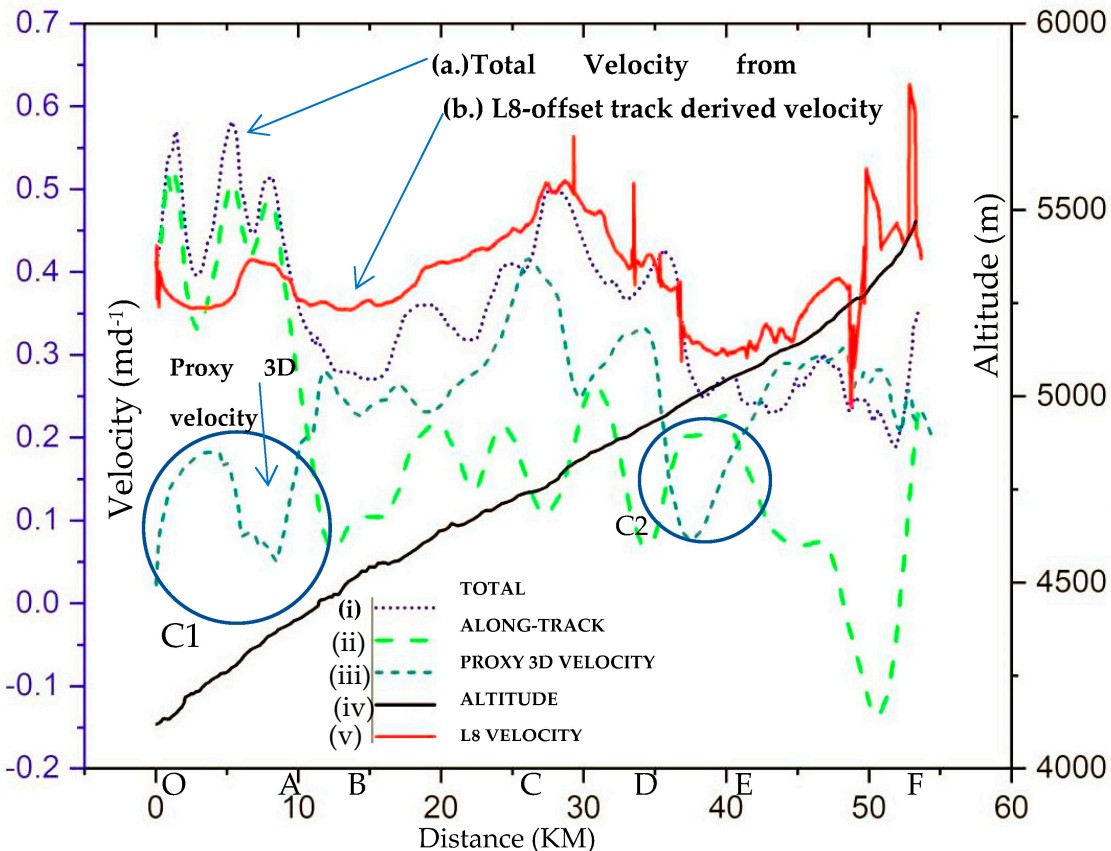

**Figure 8.** Velocity in md$^{-1}$ variation along the central line (OF) of glacier as plot (i) 3-D velocity-derived using ADIMAI, (ii) Descending pass MAI, (iii) Proxy 3-D ([1], (iv) Altitude variation meter and (v) Offset tracking-based derived from velocity using Landsat-8.

In the lower portion of the Siachen Glacier, the unwrapping error was high; hence, the terminus front was excluded from the analysis. Since section OA (of OF) in Figure 6 is almost perpendicular to the SAR ground range, the MAI component was as high as four times the ascending or descending LOS component values shown in the green profile in Figure 8. For a north/south-moving glacier, the MAI component is the dominating factor when deriving the 3-D velocity because whenever the glacier turns northward, the MAI (north/south) component shows a sudden increase, which is not captured by LOS measurements, and the proxy 3-D velocity approach as shown in the circle C1 part of the glacier.

For the further analysis of 3-D velocity from ADIMAI, the velocity variation along the three sections, OA, AC, and CE of the glacier was critically examined. Surface velocity was largest along the OA section in the terminus zone of the glacier, and maximum velocity was as much as 0.6 m/day. The contribution of ascending and descending pass velocity fields was negligible in this section, and the MAI component was the largest, as can be seen from plot (ii) Figure 8. In section AC, the velocity increased with altitude and indicated a relatively high value at points B and C in this section. Total mass flux, fed to the upper part of the Siachen glacier from adjoining glaciers, is likely to be responsible for the sudden increase in motion at points B and C. Figure 6a,b also reveals the high-value LOS movement at points B and C during both ascending and descending passes. In the CE section of the glacier, a minimum velocity of 0.25 m/day was observed. Since conventional InSAR-based observations are sensitive to only the east/west direction, in this section, the north/south component remained dominant. The upper section of the glacier showed a steady flow of 0.30 m/day.

It was observed that the 3-D velocity estimation technique, ADIMAI, showed an improvement over and combined with the ascending and descending pass approach, as stated by Jaughin et al. [1], in two ways. Firstly, ADIMAI did not make an assumption about the glacier flow directions. Overall, an increased velocity was noticed along the entire glacier compared to the two-pass approach. Secondly, due to the availability of the north/south velocity component, ADIMAI gave better results in sections where the glacier moved in a northerly direction. The real velocity profile near circles $C_1$ and $C_2$ in Figure 8 could not be measured with earlier approaches using only ascending and descending motion components. Accordingly, the technique presented here is highly recommended for precise glacier surface velocity estimation in the Himalayas and other cryospheric regions. The largest part of this glacier is the diagonal direction of radar geometry, where azimuth components are not dominant except at its terminus zone. Part of the terminus zone of this glacier flows almost parallel to the azimuth direction, and conventional across-track InSAR cannot provide velocity measurements. The accumulation zone of the glacier (section BD, Figure 6a) is the cumulative effect of ice mass fed from adjoining glaciers, which shows a sudden increase in velocity due to the increase in ice mass flux. Hence, the presented results are significant in terms of quantifying the full 3-D motion pattern of glaciers.

For more than two decades, the velocity of this glacier might have changed. For a longer-term consistent analysis, Landsat-8 data pair from 10 October 2017 to 13 December 2018 are being used to derive recent velocity trends based on subpixel offset tracking [18,19]. This optical data pair is shown in Figure 9, and its corresponding profile along the central line is shown as plot (v) in Figure 8. It has been observed that the velocity variation is similar to ADIMAI-based values except in the accumulation zone DF. For more than twenty years, the velocity of this glacier has increased in the accumulation and ablation zones (AC), but in the terminus zone, the velocity has reduced by 10 cmd$^{-1}$. L-8 results indicate that the velocity estimated using the ADIMAI technique provides a better 3-D representation than other existing methods because the north/south velocity component is incorporated in this technique.

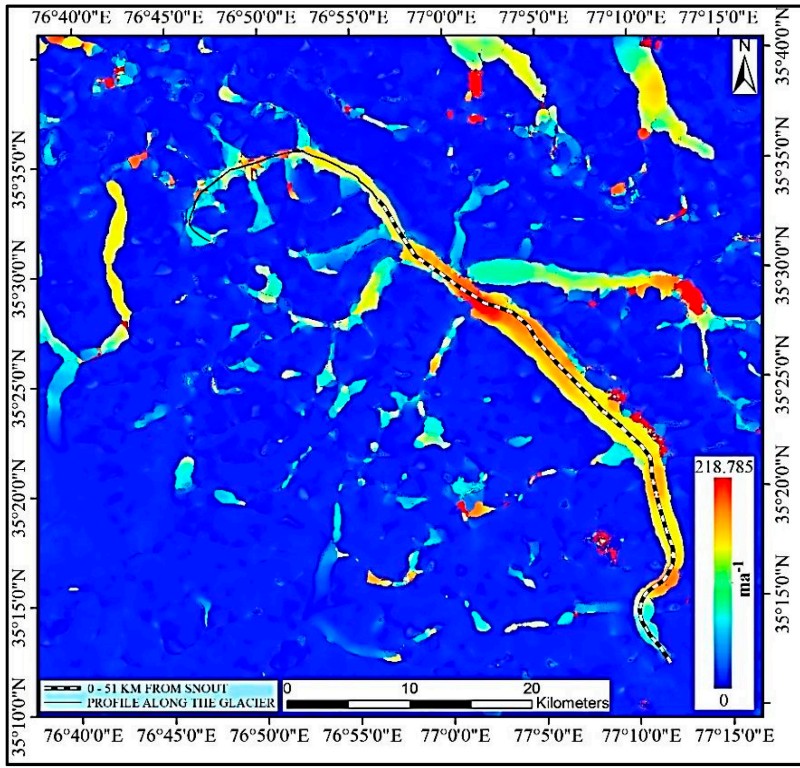

**Figure 9.** Landsat-8 velocity map during the years 2017–2018 for Siachen Glacier.

Because of arduous terrain and challenging climate conditions, ground truthing could not be conducted for accuracy assessment, and space and time-varying errors in interferometry could make the analysis complicated. Sources of error in InSAR-based deformation studies include misregistration, baseline errors, DEM errors, phase unwrapping, and atmospheric phase noise. Due to the use of precise orbit parameters, baseline errors are negligible. The interferometric phase error, due to azimuth misregistration, is proportional to the Doppler centroid, which is low for ERS-1/2 due to yaw steering. Coherence is high along the glacier; accordingly, possible phase unwrapping errors can be easily corrected.

The displacement phase is generated by subtracting the simulated phase generated using SRTM DEM from the ERS InSAR phase. Rodrigueaz et al. [32] reported that the SRTM DEM standard deviation could be more than 8.7 m.

DEM errors influence the accuracy of estimated deformation. Because of low slopes ($<5^0$) and, hence, less relative height [14] over the glaciers in the Himalayas, DEM artifacts are not prone to introduce monumental errors. Kumar et al., 2011 [21] made a detailed analysis of errors from DEM and radar geometry. For a typical ERS acquisition geometry, a terrain and baseline variation of 10 m each introduced a 0.03 mm error, which is significant in the case of the Himalayan region. The terminus zone (OA, Figure 6a) of this glacier has a large velocity component from MAI because of the high relative slope in this section due to the melting and down-wasting of glacier ice mass. A stable pixel location with a high coherence at point T (Figure 6b) outside the western boundary of the glacier has been considered a calibration point. All three movement components from InSAR and MAI were corrected with respect to the movement value at T.

The difference in atmospheric conditions between the two SAR acquisitions contributed to the differential phase. Atmospheric conditions can change even one day between ERS tandem acquisitions. Various approaches have been reported to minimise tropospheric delays. Beauducel et al. [33] proposed a methodology for atmospheric correction by analysing the correlation between the InSAR phase and local altitude. A coherence threshold was imposed to mask out the weakly correlated pixels between the two SAR acquisitions. For selected pixels of the wrapped phase, where coherence was high enough, a relationship was studied between the wrapped phase and the altitude extracted from the digital elevation model. An analysis showed that a near-linear relationship between the topography and InSAR phase existed, and systematic atmospheric error was removed from the movement signal.

To mitigate the remaining atmospheric effect due to turbulence, a multitemporal approach is necessary [34]. Hanssen [22] conducted a systematic analysis of the atmospheric effects of the interferometric phase from ERS-1/2 tandem data. It has been observed from a series of 26 ERS tandem data that rms values range from 0.5 to 4.0 radians. Luckman et al. [14] discussed, in the case of Nepal, Himalaya (the area around Everest's peak), glacier movement error due to atmospheric attenuation, baseline estimation, and DEM artifacts at around 1.7 cm/day. However, this is not necessarily valid for this study because errors due to the atmosphere are not constant across distance, and the Siachen Glacier is more than 100.0 km away from Everest's peak. For this study, available data were limited, and hence, atmospheric effects, including effects due to turbulence, could not be estimated, and this remains the most important source of error in the estimated displacement signal.

Siachen Glacier's 3-D velocity has been studied recently by combining offset tracking results from ascending and descending pass SAR pairs in conjunction with multidimensional small baseline subset technique interferometry [35,36]. It was observed that the average velocity of this glacier in 1996 was 32 cmd$^{-1}$ [21], which is insufficient due to the lack of along-track movement components. After utilising all three components in ADIMAI, the improved result was similar to [35] except in the terminus zone of the glacier. The first 10 km length of the Siachen Glacier is almost in a north-south direction, something which has not been noticed in previous studies [35,36].

## 5. Conclusions

It is demonstrated that LOS velocities from conventional ascending and descending passes InSAR can be combined with split beam MAI observations to estimate the 3-D surface velocity of Himalayan glaciers. The ADIMAI approach is not subject to the conditions of known surface slope and motion parallel to the glacier surface, as required for 3-D motion estimation from conventional InSAR. The ADIMAI technique gives improved results of surface velocity compared to Joughin et al.'s (1998) [1] approach due to the fact that MAI added the north/south movement sensitivity, which has been lacking in earlier demonstrations for polar and alpine glacier velocity estimations.

The ADIMAI technique is used to study the 3-D velocity of the Siachen Glacier's flow in the K-2 range of the Himalayas. It provides a spatially explicit measurement of glacier flow, which can be valuable in the context of understanding glacier dynamics under the influence of local geophysical conditions and climate change. It has been observed that different sections of the glacier moved at different rates, and the velocity was largest in the terminus zone of the glacier, which is not common in land-locked glaciers. In the terminus zone of the glacier, motion reaches a maximum of 0.60 m/day. The influx of ice fed to the upper-middle part of Siachen Glacier from its tributaries was responsible for the large velocity just below the meeting point of the glaciers. Except at some points, the accumulation zone of the glacier was found to be moving with a constant velocity of 0.30 m/day. Offset tracking results from the L-8 pair also provided a similar trend of glacier velocity except in the terminus zone. L-8 results give high confidence that without the NS component, accurate velocity could not be estimated. The technique presented here is highly recommended for precise glacier surface velocity estimation in the Himalayas and other cryospheric regions. MAI is limited to high InSAR coherence between acquisitions. If coherence is preserved, high accuracy can be achieved. Small temporal and geometrical baselines are ideal conditions to decipher 3-D velocity. Data from the forthcoming missions NISAR and ESA hold great promise for monitoring the velocity and further providing a proxy check to the mass balance of glaciers in the alpine Himalayas and polar regions.

**Author Contributions:** Conceptualization, V.K. and Y.L.; methodology, V.K.; software, K.A.H.; validation, V.K. and Y.L.; formal analysis, V.K.; investigation, V.K.; resources, K.A.H.; data curation, Y.L.; writing—original draft preparation, V.K.; writing—review and editing, V.K.; visualization, Y.L.; supervision, K.A.H.; project administration, K.A.H.; funding acquisition, V.K. All authors have read and agreed to the published version of the manuscript.

**Funding:** The work is supported by NMHS, MOEF and CC Govt. of India with reference no. NMHS/HF/2018-19/IF-30/08.

**Acknowledgments:** ESA has provided ERS-1/2 tandem data under Cat 1 project. GSAR SAR processing chain developed by NORCE(NORUT), Tromsø, Norway has been used for this work.

**Conflicts of Interest:** The authors declare no conflict of interest.

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
