# Peer review of "Across-Track and Multi-Aperture InSAR for 3-D Glacier Velocity Estimation of the Siachen Glacier"

_remotesensing, doi:10.3390/rs15194794_

Round 1
Reviewer 1 Report
The manuscript used across-track and Multiaperture InSAR technologies to monitor 3-D glacier velocity in the Karakoram range of Himalaya. The idea is very good, and it provides a new way to monitor the glacier changes. The manuscript can be publishable, but it can be improved in the following steps,
1) In Section 1 Introduction, few similar research about glacier velocity estimation were introduced, and it is suggested to add more references.
2) Line 48, "Earth deformation must be characterized..", “must be” is not appropriate expression.
3) Line 53-73, the basic idea of the manuscript can be described more briefly, and actually there are a lot of unnecessary words in this paragraph.
4) In section 3 Methodology, it is suggested to draw a flowchart map to describe the proposed method.
5) Line 212, Section 3.1, but there is not section title 3.2. Furthermore, the Section 3 was not easily understood, and the structure of the Section should be reorganized.
6) Line 267-403. In Section 4, there is no validations for the results, nor the comparison with other methods, please provide some validation and comparisons for the results.
Author Response
The manuscript used across-track and Multiaperture InSAR technologies to monitor 3-D glacier velocity in the Karakoram range of Himalaya. The idea is very good, and it provides a new way to monitor the glacier changes. The manuscript can be publishable, but it can be improved in the following steps,
- In Section 1 Introduction, few similar researches about glacier velocity estimation were introduced, and it is suggested to add more references.
Response: Following four more reference area added
- Liang, Q.; Wang, N. Mountain Glacier Flow Velocity Retrieval from Ascending and Descending Sentinel-1 Data Using the Offset Tracking and MSBAS Technique: A Case Study of the Siachen Glacier in Karakoram from 2017 to 2021. Remote Sens. 2023, 15, 2594. https://doi.org/10.3390/ rs15102594
- Millan, R.; Mouginot, J.; Rabatel, A.; Jeong, S.; Cusicanqui, D.; Derkacheva, A.; Chekki, M. Mapping Surface Flow Velocity of Glaciers at Regional Scale Using a Multiple Sensors Approach. Remote Sens. 2019, 11, 2498.
- Line 48, "Earth deformation must be characterized...”, “must be” is not appropriate expression.
Response: Deformation is changed to Surface Movement
- Line 53-73, the basic idea of the manuscript can be described more briefly, and actually there are a lot of unnecessary words in this paragraph.
Response: We have made this section more compact and least 100 words area removed.
- In section 3 Methodology, it is suggested to draw a flowchart map to describe the proposed method.
Res: A flowchart is introduced in the revised manuscript.
Fig.5 Flowchart for 3D displacement map
- Line 212, Section 3.1, but there is not section title 3.2. Furthermore, the Section 3 was not easily understood, and the structure of the Section should be reorganized.
Response: It is reorganized and made understandable easily
- Line 267-403. In Section 4, there is no validations for the results, nor the comparison with other methods, please provide some validation and comparisons for the results.
Response: Data has been used of 1996 so validation may be not possible but comparison has been done with critical review.
Reviewer 2 Report
Introduction
The main question was addressed by the research in the introduction.
Materials and Methods
The methods were sufficiently documented to allow replication studies. The statistical methods are valid and correctly applied.
Results and discussion
The Siachen Glacier ice-flow velocity pattern estimated using MAI technique 14 using a single InSAR pair of ERS-1/2 tandem data in comparison to conventional ascending/descending 3-D velocity estimation approach.
The results of the manuscript contribute to precise glacier surface velocity estimation in the Himalaya and other cryospheric regions as polar and alpine regions.
The quality of the figures is satisfactory, except the Figure.7.
Please, revise the layout of figure 7.
How might the findings of this study be applied to future research in the study area? For example, could this result be applied to verify the velocity trends of the Siachen glacier and changes over more than two decades and assess the factors contributing to these changes?
Conclusions
The conclusions are reporting glacier flow estimation and are supported by the results. Please, summarize about the implications of the MAI approach for precise glacier surface velocity estimation in the Himalayas and other cryospheric regions.

-Minor editing of English language required
Author Response
Reviwer-2
Comments and Suggestions for Authors
Introduction
The main question was addressed by the research in the introduction.
Response: Thank you very much.
Materials and Methods
The methods were sufficiently documented to allow replication studies. The statistical methods are valid and correctly applied.
Response: Thank you very much.
Results and discussion
The Siachen Glacier ice-flow velocity pattern estimated using MAI technique 14 using a single InSAR pair of ERS-1/2 tandem data in comparison to conventional ascending/descending 3-D velocity estimation approach.
The results of the manuscript contribute to precise glacier surface velocity estimation in the Himalaya and other cryospheric regions as polar and alpine regions.The quality of the figures is satisfactory, except the Figure.7.Please, revise the layout of figure 7.
Response: Thank you very much. We have tried to improve the quality of Figure 7 which is Figure 8. in revised version.
How might the findings of this study be applied to future research in the study area? For example, could this result be applied to verify the velocity trends of the Siachen glacier and changes over more than two decades and assess the factors contributing to these changes?
Response: Thank you very much. The study carried out related to glacier surface velocity estimation so far using SAR repeat pass imaging techniques employing InSAR can be revisited for 3D movement estimation. Since, velocity variations are proxy check for the mass balance of the glacier a better glacier mass balance analysis of Siachen glacier is possible.
Conclusions
The conclusions are reporting glacier flow estimation and are supported by the results. Please, summarize about the implications of the MAI approach for precise glacier surface velocity estimation in the Himalayas and other cryospheric regions.
MAI is limited to high InSAR coherence between the acquisitions. If coherence is preserved high accuracy can be achieved. Low temporal and geometrical baseline are ideal conditions to decipher 3D velocity.
Reviewer 3 Report
Please see attached file for full review. Thank you.

Extensive editing of English language required
Author Response
Reviwer-3
Extensive editing of English language required
Response: Entire manuscript is revisited as per suggestions given.
Regarding Figure 1. My request: Since, many satellite scene outlines are drawn over L-8 image. If I keep Legend at the centres it overlaps with other scene centres. Text annotated with boundary line of scenes. Pls consider this.
***All other comments made over the PDF version is implemented in revised version.
Round 2
Reviewer 1 Report
The reviewer's suggestion have been modified and it is have been considered for publication .
Author Response
Thank you very much
Reviewer 3 Report
Dear author
The label of Figure 1 and Figure 6 should be in the same font size, the professional figure construction is the basis of paper publishing!!!
Author Response
The label of Figure 1 and Figure 6 should be in the same font size, the professional figure construction is the basis of paper publishing!!!
It is corrected in revised manuscript.
Thank you very much for your comments and suggestions.